# β-Myrcene Mitigates Colon Inflammation by Inhibiting MAP Kinase and NF-κB Signaling Pathways

**DOI:** 10.3390/molecules27248744

**Published:** 2022-12-09

**Authors:** Saeeda Almarzooqi, Balaji Venkataraman, Vishnu Raj, Sultan Ali Abdulla Alkuwaiti, Karuna M. Das, Peter D. Collin, Thomas E. Adrian, Sandeep B. Subramanya

**Affiliations:** 1Department of Pathology, College of Medicine and Health Sciences, United Arab Emirates University, Al Ain P.O. Box 15551, United Arab Emirates; 2Department of Physiology, College of Medicine and Health Sciences, United Arab Emirates University, Al Ain P.O. Box 15551, United Arab Emirates; 3Zayed Bin Sultan Center for Health Sciences, College of Medicine and Health Sciences, United Arab Emirates University, Al Ain P.O. Box 15551, United Arab Emirates; 4Department of Radiology, Center for Health Sciences, College of Medicine and Health Sciences, United Arab Emirates University, Al Ain P.O. Box 15551, United Arab Emirates; 5Coastside Bio Resources, Deer Isle, ME 04627, USA; 6Department of Basic Medical Sciences, College of Medicine, Mohammed Bin Rashid University of Medicine and Health Sciences, Dubai P.O. Box 505055, United Arab Emirates

**Keywords:** β-myrcene, DSS colitis, inflammatory bowel diseases, MAPKs, NF-κB signaling

## Abstract

Inflammatory bowel diseases (IBDs) are chronic inflammatory disorders that include Crohn’s disease (CD) and ulcerative colitis (UC). The incidence of IBD is rising globally. However, the etiology of IBD is complex and governed by multiple factors. The current clinical treatment for IBD mainly includes steroids, biological agents and need-based surgery, based on the severity of the disease. Current drug therapy is often associated with adverse effects, which limits its use. Therefore, it necessitates the search for new drug candidates. In this pursuit, phytochemicals take the lead in the search for drug candidates to benefit from IBD treatment. β-myrcene is a natural phytochemical compound present in various plant species which possesses potent anti-inflammatory activity. Here we investigated the role of β-myrcene on colon inflammation to explore its molecular targets. We used 2% DSS colitis and TNF-α challenged HT-29 adenocarcinoma cells as in vivo and in vitro models. Our result indicated that the administration of β-myrcene in dextran sodium sulfate (DSS)-treated mice restored colon length, decreased disease activity index (DAI), myeloperoxidase (MPO) enzyme activity and suppressed proinflammatory mediators. β-myrcene administration suppressed mitogen-activated protein kinases (MAPKs) and nuclear factor-κB (NF-κB) pathways to limit inflammation. β-myrcene also suppressed mRNA expression of proinflammatory chemokines in tumor necrosis factor-α (TNF-α) challenged HT-29 adenocarcinoma cells. In conclusion, β-myrcene administration suppresses colon inflammation by inhibiting MAP kinases and NF-κB pathways.

## 1. Introduction

Inflammatory bowel disease (IBD) is a complex clinical condition associated with multifactorial etiology. The prevalence of IBD is increasing globally and imposes significant public health issues with an increasing burden on society. IBD may be classified into two major pathological subtypes: Crohn’s disease (CD) and ulcerative colitis (UC). CD is a chronic inflammatory condition that frequently presents in the later part of the small intestine, although it may affect the whole digestive tract. However, UC is considered a non-specific chronic inflammatory condition of the rectum and colon, limited primarily to the mucosa and submucosa of the large intestine. The etiology of IBD is unclear; however, it includes a predisposition to genetic susceptibility, aberrant immune response, environmental factors, gut microbial dysbiosis and mucosal dysfunction [1,2,3]. The current clinical treatment for IBD mainly includes aminosalycilates, corticosteroids, immunomodulators and surgery based on clinical presentation. Drug treatment, however, is often associated with adverse side effects that limit its use [4,5]. To illustrate, anti-TNF-α agents increase the risk of opportunistic infection and malignant tumors [6]. Similarly, patients being treated with the JAK kinase inhibitor Tofacitinib may be at risk of having a severe infection, along with the development of malignant tumors [7]. It is, therefore, imperative to search for novel treatment options for IBD. In this context, certain natural compounds and their synthetic derivatives provide a potential research opportunity in the search for new drug candidates.

β-myrcene is a well-known volatile aromatic compound commercially used as a flavor ingredient in the food and fragrance industries [8]. β-myrcene is present in various plant species, such as lemongrass oil (*Cymbopogon citratus*), rosemary (*Rosmarinus officinalis*), and is also the major component of hop and bay oils, which are used in the manufacture of alcoholic beverages [9]. β-myrcene is well known for its pharmacological effects, including analgesic, anti-inflammatory and antioxidant properties, and an antibacterial effect of β-myrcene against *Helicobacter pylori* has also been reported [10,11]. β-myrcene has shown significant anti-inflammatory and anti-catabolic effects in human chondrocyte studies (in a cell model of osteoarthritis) [12] and has a potential protective effect on UVB-induced human skin photo-aging [8]. Additionally, β-myrcene exhibited potent anti-inflammatory activity by inhibiting gastric and duodenal ulcers by enhancing gastric mucosa defense factors [13]. To date, the anti-inflammatory action of β-myrcene in colon inflammation has not been evaluated. We, therefore, investigated the effects of β-myrcene on colon inflammation using in vivo and in vitro approaches to understand the molecular targets mediating the anti-inflammatory effects.

## 2. Results

### 2.1. Effect of β-Myrcene on Disease Activity Index (DAI), Colon Length, and Myeloperoxidase (MPO) Enzyme Activity

The DAI scores were significantly higher in DSS-treated groups compared to controls. β-myrcene administration (50 and 100 mg/kg body weight) markedly decreased DAI scores in the DSS treated group. Sulfasalazine (SAZ), a clinically prescribed drug, was used as a positive control to compare the efficacy of β-myrcene. As expected, SAZ also significantly decreased DAI scores (Figure 1a). The colon length was significantly shortened in the DSS-treated group compared to controls. β-myrcene treatment markedly prevented this shortening of the colon length, compared to the DSS alone-treated group. As expected, SAZ treatment also prevented the decrease in colon length (Figure 1b,c). The level of myeloperoxidase (MPO) enzyme activity serves as a surrogate marker for neutrophil infiltration that indicates the inflammatory status of the tissue. DSS treatment significantly elevated colon MPO activity compared to controls. β-myrcene administration significantly reduced MPO activity. SAZ administration also decreased MPO activity (Figure 1d). These results indicate that β-myrcene shows a potent anti-inflammatory activity, thereby decreasing the proinflammatory effects induced by DSS treatment.

### 2.2. Effect of β-Myrcene on Colon Histology and Colon Inflammation Scores

The colonic tissues were stained using the H&E staining method to evaluate the degree of microarchitecture damage. Three different pieces from each colon were subjected to microarchitecture analyses in a blinded fashion by a clinical pathologist. The control colon histology showed well-formed histological architecture of surface villi and crypts. The submucosa and the thickness of the muscle layer were normal. DSS treatment resulted in the focal loss of surface epithelium, damaged villi with reduced villi surface and marked damage to the crypt region. The DSS treated group showed a significant level of edema in the submucosal compartment, along with thickening of the muscle layer. β-Myrcene administration protected the villi and crypt damage mediated by DSS and prevented submucosal edema. Furthermore, β-Myrcene administration also resulted in a significant reduction in colonic inflammation scores. SAZ also prevented the changes in colonic microarchitecture and reduced colon inflammation scores. (Figure 2a,b).

### 2.3. Effect of β-Myrcene on Proinflammatory Cytokines and Proinflammatory Mediators

The release of proinflammatory cytokines is the key inflammatory event in setting the inflammatory processes in IBD. Proinflammatory cytokine content in the DSS-treated colon was evaluated at the protein and mRNA levels. A significant increase in proinflammatory cytokine content in DSS-treated colon was observed. Administration of β-myrcene significantly reduced proinflammatory cytokines such as interleukin-6 (IL-6), interleukin-1β (IL-1β) and TNF-α, both at protein content and mRNA level in DSS-treated colon, indicating its potent effect in reducing key inflammatory responses (Figure 3a–f). We also investigated the effect of β-myrcene on the proinflammatory mediators cyclooxygenase-2 (COX-2) and inducible nitric oxide synthase (iNOS). DSS treatment significantly increased both COX-2 and iNOS protein expression. β-myrcene administration significantly reduced COX-2 protein expression at a higher dose (100 mg/kg bd wt). However, iNOS protein expression was reduced by β-myrcene administration at both doses. β-myrcene also reduced COX-2 and iNOS mRNA expression concomitantly in DSS-treated colon (Figure 4a–f). These results indicate that β-myrcene possesses a potent anti-inflammatory activity by limiting proinflammatory events.

### 2.4. Effect of β-Myrcene on MAPK and Nf-kB Signaling Pathways

The MAPK signaling pathways are stimulated in inflammation and activate the transcription factor NFκB which regulates the expression of many genes involved in inflammatory pathways. Proteins involved in MAPK and NFκB signaling pathways are robustly activated in the colonic mucosa of IBD patients [14,15]. We, therefore, evaluated the effect of β-myrcene on the MAPK and NF-κB signaling pathways. DSS treatment, significantly increased phosphorylation of the MAPK pathway proteins such as ERK, JNK and p38 in the colon, compared to the control group (Figure 5a–d). β-myrcene administration significantly prevented the DSS-induced phosphorylation of these proteins (Figure 5a–d). Similarly, DSS treatment increased the phosphorylation of p^ser536^NF-κB p65 protein, indicating its activation, while β-myrcene administration significantly decreased phosphorylation at the higher dose (100 mg/kg bd wt) (Figure 5d).

### 2.5. Effect of β-Myrcene on TNF-α Challenged HT-29 Colonic Adenocarcinoma Cells

TNF-α stimulated HT-29 cells are often used as an in vitro model of colon inflammation. Therefore, it provides an opportunity to evaluate the anti-inflammatory properties of β-myrcene on the human colonic cell line. Our initial experiment was to evaluate the effect of β-myrcene alone on the cell toxicity effect on the HT-29 cells. HT-29 cell monolayers were exposed to various concentrations (3.12 μM–100 μM) of β-myrcene for 24 and 48 h. β-myrcene exposure for 24 h and 48 h did not affect cell viability. Based on the cell viability data, we selected 25 and 50 μM concentrations to evaluate β-myrcene’s effect on TNF-α challenged HT-29 cells (Figure 6a,b). In these experiments, HT-29 cell monolayers were stimulated with TNF-α (1 ng/mL) for a 24 h period, and subsequently, β-myrcene was administered for 12 consecutive hours. TNF-α challenge significantly increased proinflammatory mediator, COX-2, and chemokines, CXCL-1 and IL-8 mRNA expression (Figure 6c–e). β-myrcene exposure significantly reduced the expression of COX-2, CXCL-1 and IL-8 mRNA. These results indicate that β-myrcene mitigates proinflammatory processes by limiting proinflammatory responses.

## 3. Discussion

Our results indicated that β-myrcene administration in DSS-treated animals significantly protected against histopathological changes, and reduced colon inflammation by inhibiting proinflammatory cytokine release and MAP kinases and Nf-κB signaling pathways that fuel inflammation. Furthermore, β-myrcene also decreased proinflammatory chemokine release in TNF-α stimulated HT-29 cells. These results indicate that β-myrcene possesses potent anti-inflammatory activities that are beneficial in mitigating colon inflammation.

The colonic histology reveals an increased number of infiltrating cells in the submucosa of DSS-treated mice. These infiltrating cells are the primary source of proinflammatory cytokines and increase MPO activity. The increase in MPO activity seen in DSS-treated mice is attributed particularly to granulocytes and β-myrcene administration mitigated this increase. Histological examination also reveals fewer infiltrating cells in the submucosa of the β-myrcene-administered group. Previously published studies also demonstrated that the administration of β-myrcene could prevent increased MPO activity in a rat model of ethanol-induced peptic ulcer and in acetaminophen-induced hepatotoxicity [13,16]. Our observed effects were comparable to SAZ administration. This drug is used clinically in IBD. DSS administration is known to decrease colon length due to fibrotic changes stimulated by matrix metalloproteinases (MMP-2 and MMP-9). β-myrcene administration prevented this and preserved colon length to near normal with well-formed stool pellets. These restoration effects may be attributable to the role of β-myrcene preventing MMPs from inducing the fibrotic changes. A similar fibrotic effect was inhibited by β-myrcene in a recent study conducted in an experimental heart failure model. These effects were mediated through inhibiting tissue metalloproteases, particularly MMP-2 and MMP-9 [17]. β-myrcene was also effective in preventing the increase in proinflammatory mediators (COX-2 and iNOS) and cytokines (IL-6, IL-1β and TNF-α) at protein and mRNA expression levels and resulting in colitis remission. Previously published studies also reported the anti-inflammatory effects of β-myrcene. In an in vitro cartilage degradation model of osteoarthritis, β-myrcene reduced iNOS and IL-1β and also mitigated inflammatory response in UVB-induced human skin photo-aging [8,12]. All of this experimental evidence supports that β-myrcene possesses a potent anti-inflammatory effect that can be valuable in treating inflammatory responses.

The mitogen-activated protein kinases (MAPKs) signaling pathways regulate a myriad of cellular activities. Enhanced MAPK singling pathways have been implicated in the pathogenesis of several inflammatory diseases, including inflammatory bowel disease (IBD). MAPKs are serine/threonine kinases which are important in converting extracellular stimuli into intracellular responses, leading to changes in the physiological status of the cells. The MAPKs family in mammals comprises three protein kinases: extracellular signal-regulated kinases (ERKs), the c-Jun N-terminal kinases (JNKs) and the p38 MAPKs family. MAPKs have been found to be robustly activated in IBD patients [14,15]. Therefore, it is conceivable that limiting MAPKs activation is a desirable therapeutic target [18]. β-myrcene administration inhibited MAPK signaling pathway activation, indicated by reduced phosphorylation of these proteins. Previous studies have also shown that β-myrcene exposure inhibited MAPK pathway proteins, thereby mitigating inflammation in UVB-induced human skin and in vitro osteoarthritis models [8,12]. One of the downstream target transcription factors of MAPK signaling is nuclear factor-*kappa* B (NF-κB). The NF-κB transcription factor regulates various genes responsible for the production of proinflammatory cytokines, including IL-6, IL-1, TNF-α and proinflammatory mediators such as COX-2. Therefore, limiting MAPK activation also abrogates the activation of NF-κB transcription factor to limit proinflammatory response. NF-κB transcription factor is highly activated in the inflamed gut of IBD patients. Our results indicate that β-myrcene administration in DSS-induced colitis prevented the phosphorylation of NF-κB. Therefore, β-myrcene brings anti-inflammatory action by limiting MAPK and NF-κB activation to control proinflammatory response, which is beneficial in mitigating the aberrant immune response observed in IBD.

β-myrcene is a monoterpene that is present in a large number of plant species. Various experimental studies involving β-myrcene have reported many beneficial effects, including analgesic [19], sedative [20], antioxidant [21], anti-inflammatory [22], antibacterial [23] and anticancer [24,25] effects. However, it is important to discuss the safety of β-myrcene use in preclinical and clinical settings, despite its therapeutic benefits. For example, in response to a petition from the California State Government that β-myrcene had the propensity to cause renal tubular and hepatocellular tumors in rats [26]. While previous studies had shown that β-myrcene had cytotoxic effects on cancer cells [24,26,27], this particular study investigated long-term administration of β-myrcene to rats at concentrations five or six orders of magnitude above any likely human exposure [28]. The FDA subsequently conducted a safety review of β-myrcene and stated that β-myrcene did not demonstrate genotoxic propensity and was unlikely to induce tumors in humans [26]. Despite this, β-myrcene was removed from the food additive regulations. The Joint FAO/WHO Expert Committee on Food Additives (JEFCA), declared that β-myrcene is safe to use [27].

Although the current study has identified the molecular target of the anti-inflammatory action of β-myrcene, its role in modulating colon epithelial tight junction and gut microbiota remains to be investigated.

## 4. Materials and Methods

### 4.1. Chemicals and Reagents

β-myrcene was gift from the co-author, Peter D Collin, Coastside BioResources, Maine, USA, Dextran Sulfate Sodium (DSS) purchased from (MW 36,000–50,000 kDa) MP Biomedicals (Solon, OH, USA). Hexadecyltrimethylammonium bromide (HTAB) and ortho-dianisidine dihydrochloride (ODD) were purchased from Sigma-Aldrich (St. Louis, MO, USA). IL-6, IL-1β and TNF-α ELISA kits were purchased from R&D systems (Minneapolis, MN, USA). The high-capacity reverse transcription kit was procured from Applied Biosystems (Foster City, CA, USA). Hot Firepol Evagreen qPCR Supermix from Solis Bio-Dyne (Tartu, Estonia). Macrogen Inc. (Seoul, South Korea), provided the master mix and primers for quantitative RT-PCR. The protease and phosphatase inhibitor were procured from Thermo-Scientific (Rockford, IL, USA). Antibodies were purchased from Santacruz Biotechnology (Dallas, TX, USA). All other reagents and catalog numbers are given in our previous publication [28]. HT-29 colorectal adenocarcinoma cells were obtained from the American Type Culture Collection (Manassas, VA, USA).

### 4.2. Animals

C57BL/6J male mice (~12 to 15 weeks old) weighing 25–28 g were obtained from the central animal facility, College of Medicine and Health Sciences, UAE university. The animals were housed in cages (four animals per cage) at a room temperature of 23 ± 1 °C, a 12 h light-dark cycle, with 50–60% humidity. Food and water were provided ad libitum. All studies were approved (Animal ethical application approval # ERA_2019_5927) by the Institutional Animal Care and Use Committee of the College of Medicine and Health Sciences, UAE University.

### 4.3. Experimental Design

Mice were randomly allocated to five groups. Group 1: Control, Group 2: DSS, Group 3 and 4: DSS + β-myrcene in two different doses (50 and 100 mg/kg body weight), and Group 5: DSS + Sulfasalazine (SAZ, 50 mg/kg body weight)—a standard drug that is used clinically as a positive control. The DSS (2%) was prepared freshly every day and the animals were treated for seven days (day 0 is the day of DSS administration and day 7 was animal euthanized). Mice were sacrificed using a lethal dose (100 mg/kg body wt) of pentobarbital sodium (i.p.) injection on day seven. The entire colon was resected after laparotomy. The length of each colon was measured and photographed to document colon shortening. Subsequently, for histopathology analysis, a piece of the colon at the proximal and distal colon junction was isolated and fixed in 10% formalin. The remaining colon segments were collected, snap-frozen and stored in a −80 °C deep freezer until further analysis, such as ELISA, Western blot and RT-PCR analysis.

### 4.4. Evaluation of the Clinical Score for Colitis

Mice were weighed daily and carefully observed for the presence of diarrhea and blood in stool. Scores were evaluated based on weight loss, stool type and rectal bleeding. The disease activity index (DAI) was constructed using the scoring system described previously [28].

### 4.5. Histopathological Evaluation

A fragment of colon excised at the proximal and distal part junction was fixed using 10% formaldehyde overnight. The dehydration process was carried out immersing the piece of colon in increasing ethanol concentrations and subsequently embedded in a paraffin block. Next, 2μm-thick slices from paraffin sections were stained using hematoxylin and eosin. The histological scoring of each sample was carried out by a clinical pathologist blinded to the samples. Then, the samples were scored for crypt damage, ulceration and presence of edema to obtain colon inflammation score, as described previously.

### 4.6. Myeloperoxidase Assay and ELISA

Tissue MPO activity was performed, as described previously [28,29]. The ELISA was performed to estimate proinflammatory cytokines such as IL-6, IL-1β and TNF-α levels in the colon samples, according to the manufacturer’s instructions.

### 4.7. RNA Extraction and Real-Time PCR

Total RNA from the colon was extracted using the TRIZOL (Invitrogen, Rockford, IL, USA). cDNA conversion was carried out using a high-capacity cDNA Reverse Transcription kit (Applied Biosystems, Waltham, MA, USA). Real-time polymerase chain reaction (PCR) was performed using the Quant Studio 7 Flex Real-Time PCR System (Thermo Fisher Scientific, Waltham, MA, USA) using Hot Firepol Evagreen qPCR Supermix. The 18S gene was used as the reference gene. The following conditions were applied to amplify PCR products. 95 °C for 12 min, followed by 40 cycles of 95 °C for 15 s and 60 °C for 30 s and 72 °C for 30 s. The 18S RNA was used as an internal reference gene. The gene expression changes were obtained from the CT value change that was calculated using the delta CT method (2^−∆∆CT^).

Primer sequences for each gene and cycling conditions are indicated in the following Table 1.

### 4.8. Western Blot

The colon samples were homogenized using bead tubes (Biospec, Bartlesville, OK, USA) in a Precellys homogenizer for 15 s at 6500× *g*. The homogenization protocol was repeated for five cycles. The entire homogenization process was carried out at 4 °C. The protein isolation procedure was carried out using RIPA buffer (Millipore) with protease and phosphatase inhibitor cocktail tablets (ROCHE). 20 μg of protein was extracted from different experimental conditions and resolved onto sodium dodecyl sulfate-polyacrylamide gel electrophoresis (SDS-PAGE) gels. Subsequently, these proteins were transferred to a polyvinylidene difluoride (PVDF) membrane (Thermo-Scientific, Waltham, MA, USA), as described previously [28]. The transferred proteins were immunoblotted for COX-2 (1:1000 dilution), iNOS (1:500 dilution—SIGMA), MAPKs (1:1000 dilution) (pERK/ERK, pJNK/JNK, pP38/P38) and pNF-κB/NF-κB (1:1000 dilution) pathway proteins. These proteins were normalized using GAPDH (1:10,000 dilution) (SCBT, Dallas, TX, USA) as an internal control.

### 4.9. HT-29 Cell Culture

Human colorectal carcinoma cells (HT-29 cells) were purchased from the American Type Culture Collection (Manassas, VA, USA). These cells were grown using standard cell culture conditions, as described previously [28]. Cells were sub-cultured at 80% confluency. Cells were seeded onto 96 well plates at a concentration of 5000 cells per well. After 24 h of the settling in period, these cells were incubated with various concentrations of β-myrcene (0, 3.12, 6.25, 12.5, 25, 50 and 100 µM β-myrcene MW 136.23 g/mol) for 24 and 48 h. The cell viability assay was carried out according to the manufacturer’s instructions using the Cell Titer-Glo^®^ Luminescent Cell Viability Assay kit (Promega Corporation, Madison, WI, USA). The cell viability was determined based on the amount of ATP utilized and deduced the number of metabolically viable cells in the culture. The luminescent signal was measured using the Tecan multimode plate reader system (Infinite^®^ 200 PRO, Männedorf, Switzerland), as described previously [28]. The cells were incubated with 1 ng/mL TNF-α to induce inflammatory status and subsequently exposed to β-myrcene (25 and 50 μM). The cells were harvested after the 24 h treatment protocol, and RNA was isolated, as described in the above section, to run proinflammatory chemokine gene expression analysis.

### 4.10. Statistics

Statistical analysis was carried out using GraphPad Prism (version 9) software, San Diego, CA, USA. The data obtained from different groups were compared by one-way analysis of variance (one-way ANOVA) to determine overall statistical significance. The nonparametric Tukey’s post hoc test was used for multiple comparisons. Data represented as mean ± SEM and *p*-value < 0.05 are considered statistically significant.

## 5. Conclusions

β-myrcene has potent anti-inflammatory effects by suppressing proinflammatory cytokines and mediators. These effects involve the suppression of MAP kinases and the downregulation of NF-κB pathways. This compound may be a valuable drug candidate for the treatment of colitis.

## Figures and Tables

**Figure 1 molecules-27-08744-f001:**
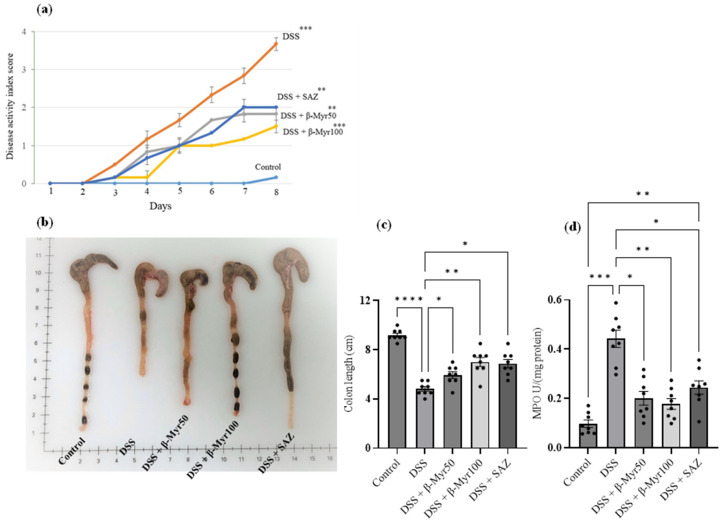
Effect of β-myrcene on DAI, colon length and MPO enzyme activity. (**a**) DSS treatment significantly increased the DAI score. Administration of β-myrcene (50 and 100 mg/kg body weight) and SAZ (clinically used drug) significantly inhibited the increase in DAI scores, compared to DSS treatment. (**b**,**c**) The mean colon length was significantly decreased in the DSS-treated group. Administration of β-myrcene and SAZ treatment significantly prevented the decrease in colon length, compared to DSS treatment alone. (**d**) MPO enzyme activity was significantly increased in the DSS treatment group, compared to untreated controls. Administration of β-myrcene and SAZ reduced MPO activity, compared to DSS treatment alone. Data were obtained from *n* = 8 animals in each group and are expressed as mean ± SEM. *p* values were obtained by one-way ANOVA followed by Tukey’s multiple comparison test. *p* ≤ 0.05 was considered statistically significant. * *p* ≤ 0.05, ** *p* ≤ 0.01, *** *p* ≤ 0.001 and **** *p* ≤ 0.0001.

**Figure 2 molecules-27-08744-f002:**
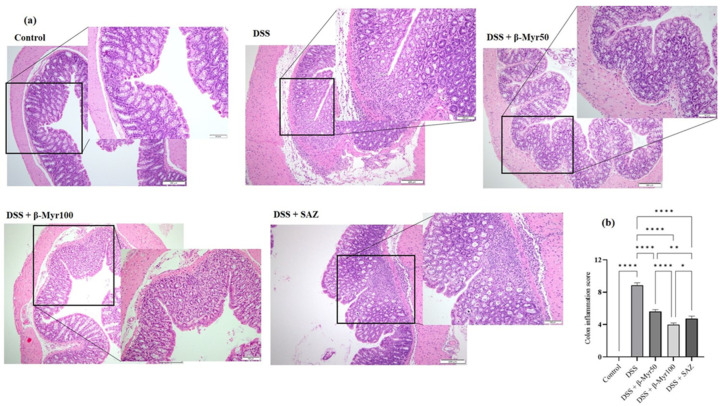
Effect of β-myrcene on colon microscopic architecture and colon inflammation scores. (**a**) Microscopic architecture showed the normal thickness of the submucosa, muscle layer, and normal crypt and villi structure in the mucosa in the control samples (Scale Bars: 100 µM). The DSS treatment resulted in the focal loss of crypts and surface epithelium, with inflammation reaching the submucosa. β-myrcene administration (50 and 100 mg/kg bd wt) in DSS-treated group significantly protected the microscopic architecture. (**b**) The colon inflammation scores were high in DSS-treated colitis group. β-myrcene administration (50 and 100 mg/kg bd wt) significantly reduced colon inflammation scores. Data were obtained from *n* = 8 animals in each group. Data expressed as mean ± SEM. *p* values were obtained by one-way ANOVA followed by Tukey’s multiple comparison test. *p* ≤ 0.05 was considered statistically significant. * *p* ≤ 0.05, ** *p* ≤ 0.01 and **** *p* ≤ 0.0001.

**Figure 3 molecules-27-08744-f003:**
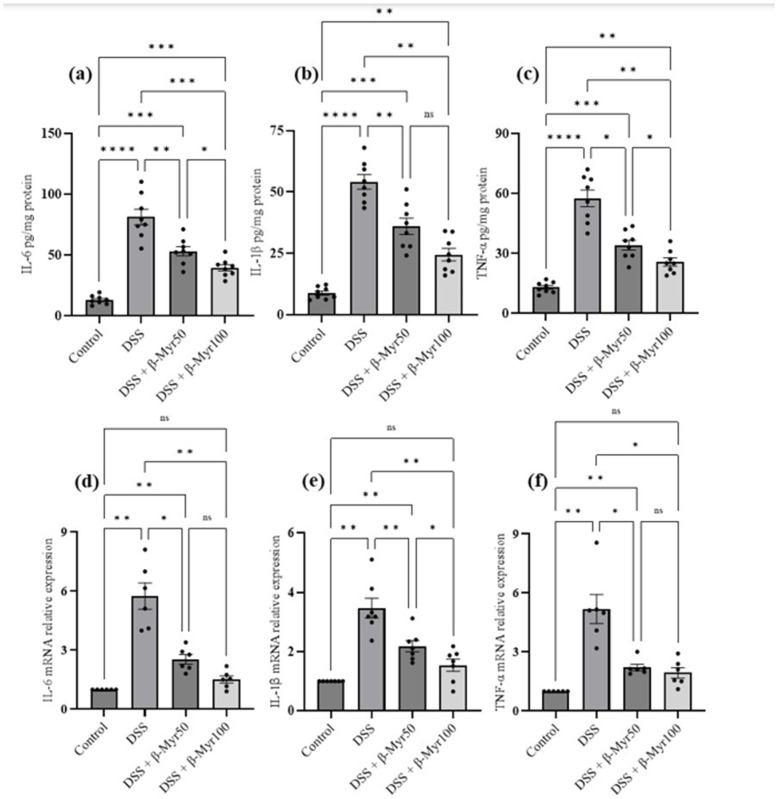
Effect of β-myrcene on proinflammatory cytokine protein and mRNA expression. DSS treatment significantly increased proinflammatory cytokine levels both at protein and mRNA levels. β-myrcene administration (50 and 100 mg/kg bd wt) significantly inhibited the DSS-treated increase in concentrations of proinflammatory cytokines, at both the protein level [IL-6 (**a**), IL-1β (**b**), and TNF-α (**c**)] and mRNA expression levels [IL-6 (**d**), IL-1β (**e**), and TNF-α (**f**)]. Data were obtained from *n* = 8 animals for ELISA and *n* = 6 animals for mRNA expression studies in each group. Data expressed as mean ± SEM. *p* values were obtained by one-way ANOVA followed by Tukey’s multiple comparison test. *p* ≤ 0.05 was considered statistically significant. * *p* ≤ 0.05, ** *p* ≤ 0.01, *** *p* ≤ 0.001 and **** *p* ≤ 0.0001 and ns (not significant).

**Figure 4 molecules-27-08744-f004:**
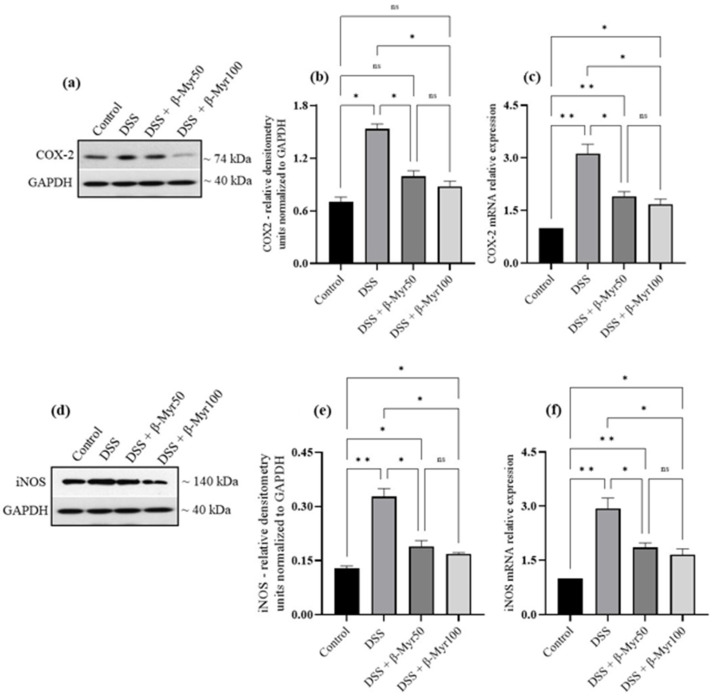
Effect of β-myrcene on proinflammatory mediators COX-2 and iNOS protein and mRNA expression. DSS treatment significantly increased COX-2 and iNOS protein and mRNA levels. β-myrcene administration (50 and 100 mg/kg bd wt) significantly prevented the DSS-treated increase in COX-2 protein (**a**,**b**) and mRNA (**c**), iNOS protein (**d**,**e**) and mRNA (**f**) levels. Data were obtained from *n* = 4 animals for Western blot analysis and *n* = 4 animals for mRNA expression studies in each group. Data expressed as mean ± SEM. *p* values were obtained by one-way ANOVA followed by Tukey’s multiple comparison test. *p* ≤ 0.05 was considered statistically significant. * *p* ≤ 0.05, ** *p* ≤ 0.01 and ns (not significant).

**Figure 5 molecules-27-08744-f005:**
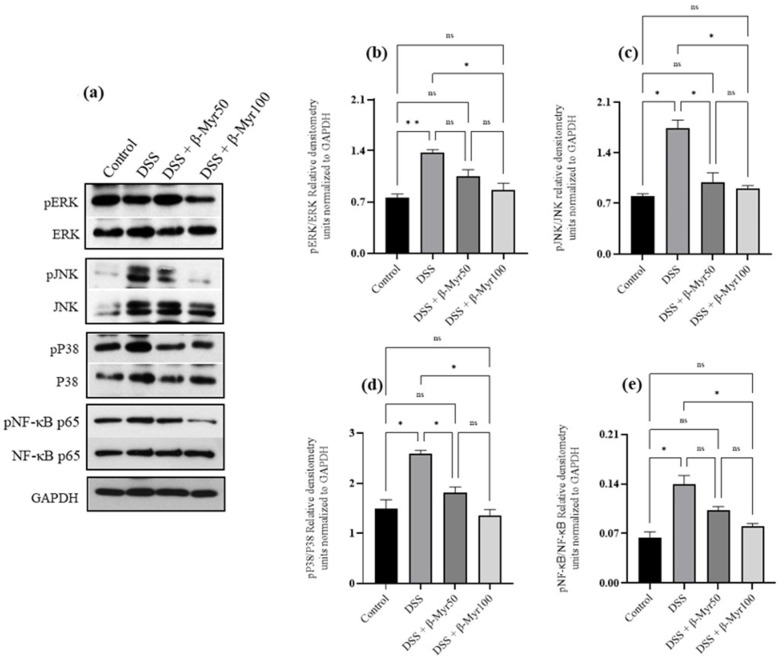
Effect of β-myrcene on MAPK and Nf-kB signaling pathway. DSS treatment significantly increased the phosphorylation of ERK, JNK, p38 and pNF-κB p65 proteins (**a**–**e**). β-myrcene administration (50 and 100 mg/kg body weight) significantly prevented the increase in the phosphorylation of these proteins, compared to DSS-treated group. Data were obtained from *n* = 4 to 5 animals for Western blot analysis. Data expressed as mean ± SEM. *p* values were obtained by one-way ANOVA followed by Tukey’s multiple comparison test. *p* ≤ 0.05 was considered statistically significant. * *p* ≤ 0.05, ** *p* ≤ 0.01 and ns (not significant).

**Figure 6 molecules-27-08744-f006:**
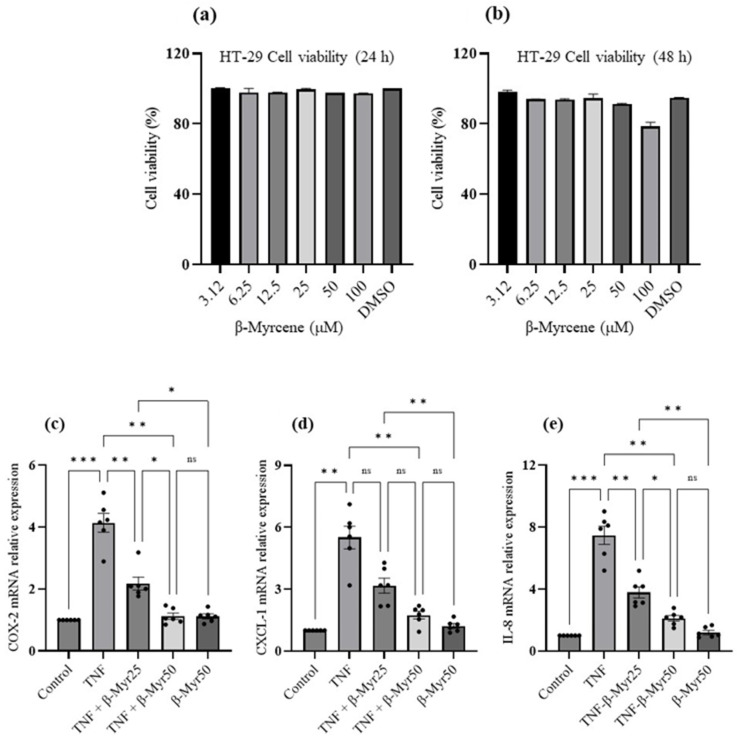
Effect of β-myrcene cell viability and on TNF-α challenged HT-29 colonic adenocarcinoma proinflammatory chemokine mRNA expression. (**a**,**b**) A dose-dependent concentration (μM) of β-myrcene was exposed (24 h and 48 h) to HT-29 cells to analyze the cell toxicity effect. β-myrcene did not affect the cell viability up to 100 μM concentration at either time (24 h and 48 h) point. HT-29 cells were challenged with TNF-α for 24 h and subsequently exposed to β-myrcene (25 and 50 μM) and proinflammatory mediator (**c**) (COX-2) and chemokine (**d**,**e**) (CXCL1 and IL-8) mRNA was analyzed. β-myrcene significantly prevented the increase in COX-2, CXCL-1 and IL-8 mRNA levels in TNF-α-treated cells. Data were obtained from *n* = 6 samples. Data expressed as mean ± SEM. *p* values were obtained by one-way ANOVA followed by Tukey’s multiple comparison test. * *p* ≤ 0.05, ** *p* ≤ 0.01, *** *p* ≤ 0.001 and ns (not significant).

**Table 1 molecules-27-08744-t001:** Primer sequence used for real-time PCR.

Gene	Forward	Reverse	PMID
Mouse IL-6	5′-TGTGTCGTGCTGTTCAGAACC-3′	5′-AGGAATCCCGCAATGATGG-3′	22326488
Mouse Il-1β	5′-TCGCTCAGGGTCACAAGAAA-3′	5′-CATCAGAGGCAAGGAGGAAAC-3′	21735552
Mouse TNF-α	5′-AGGCTGCCCCGACTACGT-3′	5′-GACTTTCTCCTGGTATGAGATAGCAAA-3′	21705622
Mouse COX2	5′-AACCGCATTGCCTCTGAAT-3′	5′-CATGTTCCAGGAGGATGGAG-3′	22158945
Mouse iNOS	5′-CGAAACGCTTCACTTCCAA-3′	5′-TGAGCCTATATTGCTGTGGCT-3′	22158945
Human COX2	5′- CAAATCCTTGCTGTTCCCACCCAT-3′	5′-GTGCACTGTGTTTGGAGTGGGTTT-3′	25810745
Human CXXL-1	5′-GCGGAAAGCTTGCCTCAATC-3′	5′-GGTCAGTTGGATTTGTCACTGT-3′	25938459
Human IL-8	5′-ACTGAGAGTGATTGAGAGTGGAC-3′	5′-AACCCTCTGCACCCAGTTTTC-3′	31273598

## Data Availability

The data presented in this study are available on request from the corresponding author.

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
