# Peer review of "β-Myrcene Mitigates Colon Inflammation by Inhibiting MAP Kinase and NF-κB Signaling Pathways"

_molecules, 2022, doi:10.3390/molecules27248744_

Round 1
Reviewer 1 Report
Review for the Manuscript β-Myrcene Mitigates Colon Inflammation by Inhibiting 2 MAPKinase Signaling Pathways.
Dear authors, thank you very much for the invitation to review this interesting manuscript.
TITLE
I suggest modifications in the title or in the Introduction. The title of an article must be consistent with the introduction and objective. In this manuscript, the title speaks of "Colon Inflammation" but not specifically of Inflammatory Bowel Disease. Therefore, I suggest changing the title or even adapting the abstract and introduction so that the focus is colon inflammation and not exactly IBD.
ABSTRACT
Please, use italics for in vitro and in vivo in lines 30-31.
Please, define DSS colitis and TNF-α in line 30. The same for NF-κB in line 34.
KEY-WORDS
Please use Inflammatory Bowel Disease instead of IBD.
INTRODUCTION
In line 44 we can read “…Crohn’s disease (CD) and ulcerative colitis (U.C.)”. Please, use “C.D. and U.C.” or “CD and UC”. Correct this along with the text. For example, in line 46.
In line 50-51, we can read “…The current clinical treatment for IBD mainly includes drugs and surgery based on clinical presentation…”
Please, include the most used drugs, for example, corticosteroids.
In lines 59-73, the authors report the effects of β-myrcene under different conditions. If there is no other work that has shown this compound in IBD, I think it should be written here. If that happens, the authors can even say that this is the first time that this biocompound is used in an IBD model.
Please, see above my comments about title and Introduction/ Abstract.
RESULTS
Most of the figures are blurred. See Figure 1, Figure 3-5. I suggest improving the quality.
Please define COX-2 and iNOS the first time they appear in the text. The same for IL, TNF-alpha and other acronyms along with the text. MAPK is defined in the abstract. However, it appears many times in the text (as an example, please, see line 161 and 164) but it is defined only in the line 240. Please, define the first time it appears in the text (not in the Abstract).
DISCUSSION
I do not think it is necessary to keep this sentence “In the current study, we investigated the role of β-myrcene using both in vivo and in vitro models of colon inflammation” in the beginning of this section.
Unlike the rest of the text, references cited in line 270 are in bold.
In line 263 we see “β-myrcene is a monoterpene that is present in a large number of plant species Various…”. Please, modify for “β-myrcene is a monoterpene that is present in a large number of plant species. Various…”
In line 269-270 we see “…While previous studies had shown that β-myrcene had cytotoxic effects on cancer cells [24,26,27], This particular…”. Please correct for ““…While previous studies had shown that β-myrcene had cytotoxic effects on cancer cells [24,26,27]. This particular…”.
Please, indicate the limitations of the study.
METHODS
Please include the year the study was approved by the Ethics Committee.
I did not fin the definition for DSS along with the text. Please, define in the Abstract (line 31), and the first time it appears in the text (line 77).
CONCLUSION
In this section, we can read “In the present study, we investigated the effects of β-myrcene on colon inflammation using in vivo and in vitro approaches. β-myrcene has potent anti-inflammatory effects by suppressing proinflammatory cytokines and mediators. The potent anti-inflammatory effects mediated by β-myrcene involve the suppression of MAP kinases and downregulation of NF-κB pathways. β-myrcene may be a valuable drug candidate for the treatment of colitis.”
The first sentence is not necessary since it was explained before. In the rest of the short paragraph, the word “β-myrcene” was repeated 3 times. I suggest replacing for:
“β-myrcene has potent anti-inflammatory effects by suppressing proinflammatory cytokines and mediators. These effects involve the suppression of MAP kinases and downregulation of NF-κB pathways. For these reasons, this compound may be a valuable drug candidate for the treatment of colitis.”
Author Response
Response to reviewer comments
Manuscript ID: molecules-2026684
Type of manuscript: Article
Title: β-Myrcene mitigates colon inflammation by inhibiting MAPKinase signaling pathways.
Authors: Saeeda Almarzooqi, Balaji Venkataraman, Vishnu Raj, Sultan Ali Abdulla Alkuwaiti, Karuna M Das, Peter D Collin, Thomas E Adrian, Sandeep B Subramanya *
Reviewer 1:
TITLE
I suggest modifications in the title or in the Introduction. The title of an article must be consistent with the introduction and objective. In this manuscript, the title speaks of "Colon Inflammation" but not specifically of Inflammatory Bowel Disease. Therefore, I suggest changing the title or even adapting the abstract and introduction so that the focus is colon inflammation and not exactly IBD.
Author’s response: Now we have changed the title to “β-myrcene mitigates Dextran Sodium Sulfate-induced Colitis by inhibiting MAP Kinase and NF-κB pathways.”
ABSTRACT
Please, use italics for in vitro and in vivo in lines 30-31.
Please, define DSS colitis and TNF-α in line 30. The same for NF-κB in line 34.
Author’s response: We have incorporated these changes into the manuscript
KEY-WORDS
Please use Inflammatory Bowel Disease instead of IBD.
Author’s response: We have incorporated the suggestion into the manuscript
INTRODUCTION
In line 44 we can read “…Crohn’s disease (CD) and ulcerative colitis (U.C.)”. Please, use “C.D. and U.C.” or “CD and UC”. Correct this along with the text. For example, in line 46.
Author’s response: We have incorporated the suggestions into the manuscript
In line 50-51, we can read “…The current clinical treatment for IBD mainly includes drugs and surgery based on clinical presentation…”
Please, include the most used drugs, for example, corticosteroids.
Author’s response: We have incorporated the suggestions into the manuscript
In lines 59-73, the authors report the effects of β-myrcene under different conditions. If there is no other work that has shown this compound in IBD, I think it should be written here. If that happens, the authors can even say that this is the first time that this biocompound is used in an IBD model.
Please, see above my comments about title and Introduction/ Abstract.
Author’s response: The following paragraph is incorporated in the manuscript (line now it is 115-118)
To date, the anti-inflammatory action of β-myrcene in colon inflammation has not been evaluated. We, therefore, investigated the effects of β-myrcene on colon inflammation using in vivo and in vitro approaches to understand the molecular targets mediating the anti-inflammatory effects.
RESULTS
Most of the figures are blurred. See Figure 1, Figure 3-5. I suggest improving the quality.
Please define COX-2 and iNOS the first time they appear in the text. The same for IL, TNF-alpha and other acronyms along with the text. MAPK is defined in the abstract. However, it appears many times in the text (as an example, please, see line 161 and 164) but it is defined only in the line 240. Please, define the first time it appears in the text (not in the Abstract).
Author’s response: These are the best possible resolution pictures we have. These figures are incorporated as JPEG pictures.
We have incorporated these suggestions into the manuscript
DISCUSSION
I do not think it is necessary to keep this sentence “In the current study, we investigated the role of β-myrcene using both in vivo and in vitro models of colon inflammation” in the beginning of this section.
Unlike the rest of the text, references cited in line 270 are in bold.
In line 263 we see “β-myrcene is a monoterpene that is present in a large number of plant species Various…”. Please, modify for “β-myrcene is a monoterpene that is present in a large number of plant species. Various…”
In line 269-270 we see “…While previous studies had shown that β-myrcene had cytotoxic effects on cancer cells [24,26,27], This particular…”. Please correct for ““…While previous studies had shown that β-myrcene had cytotoxic effects on cancer cells [24,26,27]. This particular…”.
Author’s response: We have incorporated these suggestions in the manuscript
Please, indicate the limitations of the study
Author’s response:
Although the current study has identified the molecular target of the anti-inflammatory action of β-myrcene; however, its role in modulating colon epithelial tight junction and gut microbiota ramins to be investigated.
METHODS
Please include the year the study was approved by the Ethics Committee.
I did not fin the definition for DSS along with the text. Please, define in the Abstract (line 31), and the first time it appears in the text (line 77).
Author’s response: Animal ethical application approval # ERA_2019_5927, in that 2019, indicate the year of approval.
CONCLUSION
In this section, we can read “In the present study, we investigated the effects of β-myrcene on colon inflammation using in vivo and in vitro approaches. β-myrcene has potent anti-inflammatory effects by suppressing proinflammatory cytokines and mediators. The potent anti-inflammatory effects mediated by β-myrcene involve the suppression of MAP kinases and downregulation of NF-κB pathways. β-myrcene may be a valuable drug candidate for the treatment of colitis.”
The first sentence is not necessary since it was explained before. In the rest of the short paragraph, the word “β-myrcene” was repeated 3 times. I suggest replacing for:
“β-myrcene has potent anti-inflammatory effects by suppressing proinflammatory cytokines and mediators. These effects involve the suppression of MAP kinases and the downregulation of NF-κB pathways. For these reasons, this compound may be a valuable drug candidate for the treatment of colitis.”
Author’s response: We have incorporated the suggestion.
We sincerely thank the reviewer for spending valuable time reviewing our manuscript. We agree that all these suggestions are enhanced the readability of the manuscript.

Author Response
Response to reviewer comments
Manuscript ID: molecules-2026684
Type of manuscript: Article
Title: β-Myrcene mitigates colon inflammation by inhibiting MAPKinase signaling pathways.
Authors: Saeeda Almarzooqi, Balaji Venkataraman, Vishnu Raj, Sultan Ali Abdulla Alkuwaiti, Karuna M Das, Peter D Collin, Thomas E Adrian, Sandeep B Subramanya *
Reviewer 2:
A manuscript by Saeeda Almarzooqi and coworkers entitled “β-Myrcene Mitigates Colon
Inflammation by Inhibiting MAPKinase Signaling Pathways” presents a very interesting study (based on in vivo as well as in vitro models) showing the potential use of β-Myrcene in treatment of colitis. β-myrcene – an olefinic, acyclic unsubstituted monoterpene, occurs naturally in essential oils of a large number of plant species. It is well known in brewing, because it is one of the most potent aromatic flavour components of hop essential oils, but this compound also contributes significantly to cannabis aromas, and may function analogously to the endocannabinoid system. Growing body of evidence proves biological activities of β-myrcene, including analgesic, sedative, antidiabetic, antioxidant, anti-inflammatory, antibacterial and anticancer effects. The study by Almarzooqi andcoworkers focuses on the anti-inflammatory activity of β-myrcene in the context of colitis treatment. The study was well planned and correctly executed. The manuscript presents many interesting results from the in vivo part of research, based on murine model of DSS-induced colitis. To investigate signaling pathways linked to inflammatory reaction of intestinal epithelium the Authors used a popular in vitro model of HT29 cell line, which is a human colon cancer-derived cell line.
I don’t have major remarks regarding methods used or interpretation of the results. However, in my opinion the manuscript needs further English language editing before publication. The abstract needs
major revision – it is unpleasant to read, especially when compared with the rest of the manuscript.
Sentences, such as:
- Inflammatory bowel disease (IBD) comprised of Crohn disease and ulcerative colitis is rising
globally.
- The natural substance of plant origin takes a significant lead in the search for
pharmacologically active substances that could be used for IBD treatment.
- β-myrcene administration suppressed mitogen-activated protein kinase (MAPKs) and NF-κB
signaling pathways, which are critical in fueling inflammation.
need careful editing (especially in fragments marked in bolded font. Authors have to remember that they were analyzing several MAP kinases, so they should use a plural form (not singular as it is used in lines 33-34) and the title should be corrected as a space is needed between MAP and kinase (The correct title should be: “β-Myrcene Mitigates Colon Inflammation by Inhibiting MAP Kinase Signaling Pathways”
Conclusion in the abstract should be more specific – the sentence: In conclusion, β-myrcene
administration suppresses colon inflammation does not really conclude the mechanism of action of β-Myrcene.
Author’s response: We agree with the reviewer that the abstract does not read well. We have worked on the abstract and the modified version of the abstract now as follows.
Abstract: Inflammatory bowel diseases (IBDs) are chronic inflammatory disorders that include Crohn's disease (CD) and ulcerative colitis (UC). The incidence of IBD is rising globally. However, the etiology of IBD is complex and governed by multiple factors. The current clinical treatment for IBD mainly includes steroids, biological agents, and need-based surgery based on the severity of the disease. Current drug therapy is often associated with adverse effects, which limits its use. Therefore, it necessitates the search for new drug candidates. In this pursuit, phytochemicals take the lead in the search for drug candidates to benefit from IBD treatment. β-myrcene is a natural phytochemical compound present in various plant species which possesses potent anti-inflammatory activity. Here we investigated the role of β-myrcene on colon inflammation to explore its molecular targets. We used 2% DSS colitis, and TNF-α challenged HT-29 adenocarcinoma cells as in vivo and in vitro models. Our result indicated that the administration of β-myrcene in dextran sodium sulfate (DSS)-treated mice restored colon length, decreased disease activity index (DAI), myeloperoxidase (MPO) enzyme activity, and suppressed proinflammatory mediators. β-myrcene administration suppressed mitogen-activated protein kinases (MAPKs) and nuclear factor-κB (NF-κB) pathways to limit inflammation. β-myrcene also suppressed mRNA expression of proinflammatory chemokines in tumor necrosis factor-α (TNF-α) challenged HT-29 adenocarcinoma cells. In conclusion, β-myrcene administration suppresses colon inflammation by inhibiting MAP kinases and NF-κB pathways.
Further mistakes in the text that need to be corrected:
- Lines 59-60: β-myrcene a well-known volatile aromatic compound commercially used as a flavor ingredient in the food and fragrance industries. – this sentence is missing a verb. Add “is” to this
sentence: β-myrcene is a well-known volatile aromatic compound commercially used as a flavor
ingredient in the food and fragrance industries.
Author’s response: We have incorporated it into the manuscript
- Lines 130-133: Administration of β-myrcene significantly reduced proinflammatory cytokines
such as IL-6, IL-1β, and TNF-α both at protein content and mRNA level was observed in DSS-treated colon, indicating its potent effect in reducing key inflammatory response (Fig 3a-3f).
(remove was observed).
Author’s response: We removed ‘was observed.’
- Line 163 – correct the word: genes (now it is gtnes).
Author’s response: We have corrected it.
- Lines 189-190: In these experiments, HT-29 cell monolayers were stimulated with TNF-α
(1ng/mL) for 24h period, and subsequently, β-myrcene was exposed for 12 h. Change the last
part of this sentence to: … and subsequently, β-myrcene was administered for 12 consecutive
hours.
Author’s response: We have corrected it.
- In chapter 4.1. of Material and Methods – do not use the word procured! Replace it with: was
purchased or was supplied by.
Author’s response: We have corrected it.
- Line 302: … DSS + Sulfasalazine (SAZ, 50mg/kg body weight) a standard drug that used
clinically… – a verb is missing. Add “is” to this sentence: … DSS + Sulfasalazine (SAZ, 50mg/kg
body weight) a standard drug that is used clinically…
-
Author’s response: We have corrected it.
Chapter 4.5, line 315: replace the phrase: A piece of colon… with A fragment of colon …
- Correct the format of Table presented in chapter 4.6 – add a number of this table (e.g. Table 1), change the font of primer sequences so that the entire sequences are in one line (from 5’ to 3’ end), not split into two lines and centered!
Author’s response: We have corrected it.
- Chapter 4.10 - specify which ANOVA was used in statistical analyses – One way ANOVA?
Authors response: We used One way ANOVA and the same is corrected in the manuscript.
We sincerely thank the reviewer for the careful attention given to the details that were missing from our manuscript. We agree that these changes have significantly enhanced the readability of the manuscript.
